# Quantitative MRI of Pancreatic Cystic Lesions: A New Diagnostic Approach

**DOI:** 10.3390/healthcare10061039

**Published:** 2022-06-02

**Authors:** Paul Andrei Ștefan, Roxana Adelina Lupean, Andrei Lebovici, Csaba Csutak, Carmen Bianca Crivii, Iulian Opincariu, Cosmin Caraiani

**Affiliations:** 1Anatomy and Embryology, Morphological Sciences Department, “Iuliu Hațieganu” University of Medicine and Pharmacy, Victor Babes, Street, Number 8, 400012 Cluj-Napoca, Romania; stefan_paul@ymail.com (P.A.Ș.); bianca.crivii@umfcluj.ro (C.B.C.); iulian_opincariu@hotmail.com (I.O.); 2Radiology and Imaging Department, County Emergency Hospital, Cluj-Napoca, Clinicilor Street, Number 5, 400006 Cluj-Napoca, Romania; andrei1079@yahoo.com (A.L.); csutakcsaba@yahoo.com (C.C.); 3Histology, Morphological Sciences Department, “Iuliu Hațieganu” University of Medicine and Pharmacy, Louis Pasteur Street, Number 4, 400349 Cluj-Napoca, Romania; 4Radiology, Surgical Specialties Department, “Iuliu Hatieganu” University of Medicine and Pharmacy, Clinicilor Street, Number 3–5, 400006 Cluj-Napoca, Romania; 5Department of Medical Imaging, “Iuliu Hațieganu” University of Medicine and Pharmacy Cluj-Napoca, 400012 Cluj-Napoca, Romania; ccaraiani@yahoo.com; 6Department of Radiology, Regional Institute of Gastroenterology and Hepatology “Prof. Dr. Octavian Fodor”, 400158 Cluj-Napoca, Romania

**Keywords:** ADC, artificial intelligence, IMPN, MRI, pancreas, pancreatic cyst, radiomics, serous cystadenoma, texture analysis

## Abstract

The commonly used magnetic resonance (MRI) criteria can be insufficient for discriminating mucinous from non-mucinous pancreatic cystic lesions (PCLs). The histological differences between PCLs’ fluid composition may be reflected in MRI images, but cannot be assessed by visual evaluation alone. We investigate whether additional MRI quantitative parameters such as signal intensity measurements (SIMs) and radiomics texture analysis (TA) can aid the differentiation between mucinous and non-mucinous PCLs. Fifty-nine PCLs (mucinous, *n* = 24; non-mucinous, *n* = 35) are retrospectively included. The SIMs were performed by two radiologists on T2 and diffusion-weighted images (T2WI and DWI) and apparent diffusion coefficient (ADC) maps. A total of 550 radiomic features were extracted from the T2WI and ADC maps of every lesion. The SIMs and TA features were compared between entities using univariate, receiver-operating, and multivariate analysis. The SIM analysis showed no statistically significant differences between the two groups (*p* = 0.69, 0.21–0.43, and 0.98 for T2, DWI, and ADC, respectively). Mucinous and non-mucinous PLCs were successfully discriminated by both T2-based (83.2–100% sensitivity and 69.3–96.2% specificity) and ADC-based (40–85% sensitivity and 60–96.67% specificity) radiomic features. SIMs cannot reliably discriminate between PCLs. Radiomics have the potential to augment the common MRI diagnosis of PLCs by providing quantitative and reproducible imaging features, but validation is required by further studies.

## 1. Introduction

In recent years, the widespread use of abdominal high-resolution cross-sectional imaging for various indications has led to the increased detection of pancreatic cystic lesions (PCLs) [1]. Histologically, PCLs are classified as neoplastic and non-neoplastic. Pseudocysts account for two-thirds of non-neoplastic PCLs, but this category includes a wide variety of other, lesser-known entities (lymphoepithelial, benign epithelial, retention, and congenital cysts) [2]. The incidence of pancreatic cystic neoplasms is highly variable, mostly dependent on the imaging detection method: 2.2–2.6% with computed tomography (CT) [3] and 14–45% with magnetic resonance imaging (MRI) [4]. Based on the contained epithelium, neoplastic cysts are broadly categorized as serous or mucinous. Mucinous neoplastic pancreatic cysts (MNPCs) include few entities (mucinous cystic neoplasms (MCNs) and intraductal papillary mucinous neoplasms (IPMNs)), while non-mucinous neoplastic pancreatic cysts (nMNPCs) (often erroneously called serous pancreatic neoplasms) include a wider variety of lesions (serous cystadenomas (SCA), solid pseudopapillary neoplasms, cystic pancreatic neuroendocrine tumors, cystic pancreatic ductal adenocarcinomas, and other, less often encountered malignancies). 

The imaging distinction between mucinous and non-mucinous PCLs is crucial because MNPCs have malignant potential and therefore require close follow-up [5], while some nMNPCs, such as SCAs, have a very reduced risk of malignant transformation and do not require intervention unless symptomatic [5,6]. Although pre-operative imaging can highlight the “worrisome” signs of PLCs’ neoplastic transformation, distinguishing between different PCL subtypes (especially MNOCs from nMNPCs) is frequently not possible [7]. Often, PCLs present with non-specific and overlapping imaging characteristics, which become even less specific with the decreasing size of the observation [8]. For these reasons, the correct imaging identification of the PCL subtype is limited, which leads to their misclassification in up to 50% of their encounters [9]. Considering these limitations, a definite pre-operative imaging diagnosis of the PLC subtype is rarely possible. Therefore, if imaging could offer more confident discrimination between mucinous and non-mucinous PCLs, it would facilitate the subsequent management of these patients. Pathology remains the gold standard in the final diagnosis of PCLs. Although this method can provide almost 100% specificity in detecting malignant cysts, and can successfully discriminate serous from mucinous fluid contents, the fluid sampling is often inadequate and requires invasive procedures, which expose the patients to a series of risks [10]. Therefore, the need to evaluate new imaging features that can increase the pre-operative non-invasive diagnostic of MNPCs, preferably through non-subjective, quantitative, and reproducible parameters.

The fluid contents of PCLs exhibit specific characteristics for different histopathological groups in terms of cellularity and physical and biochemical properties [2,11,12,13,14]. At cytological analysis, the aspirates of MCPNs contain extracellular mucin [2,12], a few sparse cell populations, and columnar mucinous sheets [2,12]. However, fluid analysis has a limited role in distinguishing between MCPN entities, as they share many cytomorphologic features [2]. On the other hand, the fluid of SCAs is sparsely cellular, having only a few fragments of flat sheets and/or loose clusters of cuboidal cells with glycogenated cytoplasm and indistinct cytoplasmic borders [2]. Pseudocysts and other non-neoplastic cysts contain high levels of amylase and other pancreatic enzymes [15]. Aspirates from pseudocysts are typically paucicellular and consist of granular debris, hemosiderin-laden macrophages, and bile [2]. The biochemical and physical fluid properties can also aid the differentiation between MNPCs and nMNPCs. Tumor markers such as CA 19-9, CEA, and CA 72-4 have a significantly higher concentration in MNPCs’ fluid compared to other cystic entities [14]. The fluid viscosity is usually lower in pseudocysts and serous cystadenomas when compared with mucinous cystadenoma and mucinous cystadenocarcinoma [13]. This increased fluid viscosity was also demonstrated to be associated with malignant or potentially malignant cysts [14].

It is believed that certain microscopic tissue characteristics can be reflected in medical images, producing alterations that can be too subtle to be perceived by the human eye. A radiomics branch, namely texture analysis (TA), represents an imaging processing technique that involves the extraction of specific parameters that reflect the pixel intensity and variation patterns, thus providing a quantitative and detailed description of the image contents [16]. In recent years, TA has been integrated as a core component of computer-aided discrimination (CAD) applications that aim to increase the confidence in imaging detection and the characterization, especially of oncological pathologies [17]. Up to this moment, more than a dozen TA studies involving pancreatic cysts have been performed [18,19,20,21,22,23,24,25,26,27,28,29,30]. To the best of our knowledge, all of these previous reports only evaluated the radiomic features based on CT images alone [19,20,21,22,23,24,25,26,27,28,29,30], while the contribution of MRI-derived TA parameters has been neglected.

Diffusion-weighted imaging (DWI) is an MRI technique that reflects the Brownian motions of the water in tissues. This motion degree can be assessed qualitatively and quantitatively by the apparent diffusion coefficient (ADC) maps [31] and is widely available in most clinical practices. The role of ADC measurements in differentiating between mucinous and serous PLCs has been investigated before, with variable and often contradicting results [32,33,34,35,36]. It would have been expected for mucinous tumors to display lower ADC values due to the dense mucin content, but very often, a higher degree of restriction was observed in serous-containing lesions [32,33,34,35,36,37]. Similarly, the same density difference would be expected to produce alterations in the T2 signal intensities (SIs) of PCLs, but, to the best of our knowledge, the role of quantitative T2 SIs in differentiating between mucinous and serous PLCs has never been investigated before. 

We hypothesized that mucinous pancreatic cysts would have distinct MRI signal characteristics compared to non-mucinous pancreatic cysts (due to their very different content), but these differences cannot be observed and classified by the human eye. Therefore, we proposed two methods of quantitative signal evaluation based on the signal intensity measurements (SIMs) of PLCs on ADC maps and T2 and DWI sequences, and the radiomics/texture analysis on the PLCs based on T2WI and ADC maps.

## 2. Materials and Methods

### 2.1. Study Group

This Health Insurance Portability and Accountability Act-compliant, single-institution, retrospective pilot study was approved by the institutional review board, and informed consent was waived due to the retrospective nature of this research. We aimed to include patients with PLCs who underwent MRI examinations in our institution from August 2017 to April 2019, with the possibility of future histopathological confirmation when biopsy/surgery was suggested, and/or subsequent imaging examinations every 6 months for at least 12 months in cases of typically benign lesions that were not biopsied or operated on. The patient selection was performed by one researcher (CC, a radiologist specialized in abdominal MRI) who was aware of both the MRI images and the patients’ clinical data. The researcher was not involved in subsequent quantitative image analysis workflow to reduce the possibility of potential bias.

Firstly, a keyword search (using the terms “pancreatic + cyst/cystic/cystic lesions”, alternative, and combined) was conducted in the imaging database of our institution to identify MRI examinations corresponding to PCLs. The search resulted in 181 image reports. Each report was analyzed and the researcher excluded the ones that were not referring to PCLs (*n* = 28) or when the PCLs reports mentioned a lesion diameter of less than 10 mm (*n* = 23). Secondly, the medical records of the patients with the remaining 130 examinations were retrieved from the archive of our healthcare institution and investigated for disease-related data. Patients that were transferred to another institution for follow-up or treatment (*n* = 4) were excluded. Patients without a final diagnosis (established by clinical/imaging data or histopathological analysis) were eliminated (*n* = 49). When patients had multiple MRI examinations, the “reference” examination was considered the one performed before the surgery/fine needle aspiration (FNA) (for pathologically confirmed lesions), and the most recent study for patients that underwent imaging and clinical follow-up. The duplicates were then eliminated (*n* = 9). Afterward, the MRI examinations were reevaluated. When multiple lesions were detected in the same patient, the researcher only marked the ones that had a final diagnosis. In addition, all lesions were re-measured and the ones that did not meet the size criterion were excluded from further evaluation (*n* = 2). All examinations in which the T2-weighted sequences (T2WI), the DWI, or the ADC maps were affected by artifacts were excluded from the further investigation (*n* = 12). Finally, 59 lesions from 54 patients were included in the study group.

### 2.2. Image Acquisition

All MRI examinations were performed on the same machine (General Electric Optima 360MR Advance system, Waukesha, WI, USA; 1.5 Tesla). Dedicated array coils were used to cover the abdominal and pelvic regions. The imaging protocol varied because the examinations were selected from a range of 3 years, but each examination included a T2 single-shot fast spin-echo (T2 SS-FSE) and DWI sequences, which were the only sequences used for this study. Axial DWI acquisitions were synchronized with respiratory movements and computed for the same three b-values (50, 400, and 800 s/mm^2^). The DWI sequences were acquired using the same slice interval and thickness as well as location as the ones used for the standard axial sequences. The DWI parameters were: repetition time (TR), 10,000 ms; echo time (TE), 64 ms; slice thickness, 6 mm; interval, 1 mm and acquisition matrix, 128 × 128. The ADC and Exponential Apparent Diffusion Coefficient (eADC) functional maps were automatically obtained on the scanner computer, using echo-planar imaging (EPI) correction using the following parameters: confidence level, 0.9; lower threshold, 20, and kernel size, 2. The acquisition parameters for the T2 SS-FSE sequence were: TR, 1100 ms; TE 95 ms; slice thickness, 4 mm; slice spacing, 1 mm; and acquisition matrix, 256 × 152.

### 2.3. Image Interpretation

Each examination was reviewed by the same radiologist (CC). When different types of pancreatic cystic lesions were observed within the same organ, the images were cross-referenced with the pathological and ultrasonography results, and other medical data to ensure the selection of lesions that were previously documented. Selected lesions were marked. The researcher chose one slice on the T2-weighted sequence where the lesions’ fluid content was better displayed. The chosen T2-weighted sliced was synchronized with the DWI sequences and the ADC maps, and one image from every mentioned sequence was retrieved and used for subsequent analysis, after being anonymized.

### 2.4. Signal Intensity Measurements

The previously selected images were imported on a dedicated workstation (General Electric, Advantage workstation, 4.7 edition, Boston, MA, USA). The quantitative SIMs were performed in consensus by two observers (IO and AL). Both researchers were blinded to patients’ clinical history, laboratory data, and histopathology characteristics. The signal intensity values on the T2WI, the three DWI sequences (b = 50, b400, 1000 mm^2^/s), and the ADC maps were measured by placing an elliptical region of interest (ROI). The observers carefully placed the ROI within the cystic lesions, respecting the cystic walls, intra-lesional debris, or solid components. Moreover, the ROIs were drawn to avoid vascular motion and abdominal wall artifacts, as well as visible vascular and biliary structures. The minimum diameter of every ROI was set to at least 0.2 cm^2^ and was placed approximately in the same location on each sequence, using synchronized slices. Each researcher performed one set of measurements. The values were averaged and used for subsequent statistical analysis. A univariate analysis test (the Mann–Whitney U test) was conducted to compare average SIMs between the groups. A synthetic reproduction of the ROI positioning process is depicted in Figure 1. Further, the coefficient of variation (COV) from duplicate measurements was calculated using the logarithmic method to determine the reproducibility of the measurements conducted by the two radiologists, thus estimating the within-run imprecision [38]. The intraclass coefficient (Kappa, ICC) was also determined from the same data.

### 2.5. Radiomics Workflow

The radiomics approach consisted of five steps: image pre-processing, lesion segmentation, feature extraction, feature selection, and prediction. 

#### 2.5.1. Image Pre-Processing and Segmentation

The same T2WI and ADC maps used for the SIMs were imported into a dedicated software for texture analysis, QMaZda (MaZda, Institute of Electronics, Technical University of Lodz, Lodz, Poland). Subsequently, the imported image’s gray levels were normalized based on the mean and three standard deviations of gray level intensities to reduce the contrast and brightness variations (which could affect the true image textures) The image segmentation process was performed by a second researcher (RAL) who was blinded to the outcomes of the patients and was also not involved in the SIM process. On T2WIs, the researcher incorporated each lesion into a two-dimensional ROI. The first step of the ROI definition process was performed semi-automatically. The researcher placed a seed inside each cyst and the software automatically delineated the structure of interest, based on gradient and geometry coordinates. In the second step, if a complete overlap between the ROI and the structure’s contours was detected, the ROI was manually adjusted. Afterward, the defined ROI was transferred to the ADC maps. Manual adjustments were performed when an uncomplete overlapping was observed (Figure 2).

#### 2.5.2. Feature Extraction

After ROI delineation, the software automatically extracted the texture features (texture parameters) using preset computation methods (Table 1). From each lesion, a total of 550 parameters were extracted (275 parameters from T2WI and 275 parameters from ADC maps) and used for subsequent analysis.

#### 2.5.3. Feature Selection

The feature selection workflow was identical for the parameters extracted from the T2WIs as well as the ones computed from the ADC maps. Firstly, two predefined reduction techniques were applied to highlight the parameters with the highest discriminatory potential: the probability of classification error and average correlation coefficients (POE + ACC) and Fisher coefficients (F, the ratio of between-class to within-class variance), each of them providing a set of 10 texture features. The Fisher algorithm selected features with maximized differences between two groups, while the POE + ACC algorithm introduced features with high discriminatory potential and the least correlation with features that were already selected. Afterward, the absolute values of the highlighted parameters were compared between MNPCs and nMNPCs by performing a univariate analysis test (Mann–Whitney U). The statistical significance level was set at a *p*-value of below 0.05. To evaluate the reproducibility and stability of the selected texture feature sets, 24 patients were randomly selected for a double-blinded comparison and the same radiologist redefined each ROI, approximately two weeks after the initial process. Features with an intraclass correlation coefficient (ICC) lower than 0.85 were excluded from further analysis.

#### 2.5.4. Class Prediction

We investigated which of the remaining parameters could function as independent predictors for mucinous lesions (on T2WI and ADC maps, respectively). In this regard, a multiple regression analysis (using the “enter” input model) was conducted, with the computation of the variance inflation factor (VIF) and the coefficient of determination (R-squared). The “enter” input model included all variables that showed a *p*-value of below 0.05 and removed all variables that showed a *p*-value of more than 0.01. Features that showed a VIF of greater than 10^4^ were removed from further analysis, since a high VIF indicates multicollinearity. The predicted values were saved and subsequently used in a receiver-operating characteristics (ROC) analysis to assess the diagnostic power of the entire prediction model. The ROC analysis was also used to determine the diagnostic power of texture features that were associated with mucinous-containing cysts, along with the calculation of the area under the curve (AUC), sensitivity, and specificity, with 95% confidence intervals (CIs). The ROC curves were calculated using the DeLong et al. method, and the binomial exact confidence intervals for the AUCs were reported. Optimal cut-off values were chosen using a common optimization step that maximized the Youden index for predicting patients with malignancies. Sensitivity (Se) and specificity (Sp) were computed from the same data, without further adjustments. The same texture workflow was conducted for both T2- and ADC-based features. Statistical analysis was performed using commercially available dedicated software, MedCalc v14.8.1 (MedCalc Software, Mariakerke, Belgium) and SPSS Statistics for Windows, version 18.0 (SPSS Inc., Chicago, IL, USA). The radiomics workflow diagram is displayed in Figure 3.

## 3. Results

Of the 181 examinations of patients with pancreatic cysts that were performed in our department during the study period, 54 subjects (35 women and 19 men, with a mean age of 46 years; range, 32–78 years) were included in this research. The final diagnosis of their lesions was concluded through three possible scenarios (surgery, 30; clinical/follow-up, 18; FNA, 6). Nine patients had FNA followed by surgery and they were considered “surgically proven”. From the patient cohort, a total of 59 pancreatic lesions were selected. Two individual lesions were selected from three patients with pseudocysts, and two cysts from different branches were selected each from two patients with IPMNs. The cohort included the following lesions: pseudocysts (*n* = 19), walled-off necrosis (WONs; *n* = 6), SCAs (*n* = 4), solid pseudopapillary neoplasms (*n* = 2), cystic ductal adenocarcinoma (*n* = 1), simple cysts (*n* = 2), cystic pancreatic neuroendocrine tumor (*n* = 2), IPMN (total number of lesions, *n* = 18; branch-duct IPMN, *n* = 12; main-duct IPMN, *n* = 6), mucinous cystadenomas (*n* = 4), and mucinous cystadenocarcinomas (*n* = 1). The confirmation for every type of lesion was as follows: pseudocysts (clinical/imaging, *n* = 13; surgery, *n* = 4; FNA, *n* = 2), WONs (clinical/imaging, *n* = 1; surgery, *n* = 4; FNA, *n* = 1), SCAs (clinical/imaging, *n* = 3; surgery, *n* = 1), solid pseudopapillary neoplasm (surgery, *n* = 2), cystic ductal adenocarcinoma (surgery, *n* = 1), simple cyst (clinical/imaging, *n* = 1; FNA, *n* = 1) cystic pancreatic neuroendocrine tumor (surgery, *n* = 2), IPMN (surgery, *n* = 12; FNA, *n* = 1), mucinous cystadenomas (surgery, *n* = 4; FNA, *n* = 1), mucinous cystadenocarcinomas (surgery, *n* = 1). Patients’ characteristics are presented in Table 2.

The coefficient of variation of the SIMs conducted by the two radiologists on the T2WI was 9.6% (CI, 7.58–11.66%), on the b50 images was 12.26% (CI, 9.65–14.93%), on the b400 images was 16.2% (CI, 12.78–19.91%), on the b1000 was 32.38% (25.05–40.14%), and on the ADC maps was 6.91 (CI, 5.46–8.37%). The COV and ICC results are displayed in Table 3.

Based on the features extracted from T2WIs, the Fisher and POE + ACC methods highlighted two common parameters (Perc01 and RHD6GLevNonU). In total, 18 unique parameters extracted from T2WI were selected by the reduction methods. Three parameters, each representing computational variations of the Correlation feature (CV2D6Correlat, *p* = 0.968; CN1D6Correlat, *p* = 0.953; CZ1D6Correlat, 0.653) showed no statistically significant results upon univariate analysis. In addition, the RND6Fraction and the RND6ShrtREmp parameters showed ICCs below the thresholds (0.62 and 0.51 respectively). The T2-based multivariate analysis showed a coefficient of determination of 0.65, an R^2^-adjusted of 0.45, and a multiple correlation coefficient of 0.8. Five parameters (CH3D6Contrast, CH5D6InvDfMom, and three variations of Grey Level Non-Uniformity (RHD6GLevNonU, RND6GLevNonU, and RVD6GLevNonU)) were independent predictors for mucinous cysts. Their combined ability was able to identify MNPCs with a high sensitivity of 100% (95%CI, 83.2–100%) and a good sensitivity of 86.67% (95%CI, 69.3–96.2%). 

Based on the features extracted from ADC maps, the two reduction methods selected 16 unique parameters. Four parameters were highlighted by both Fisher and POE + ACC (CN3D6Contrast, CN2D6Contrast, RZD6GLevNonU, Perc90). Six parameters showed no statistically significant result upon univariate analysis (CN2D6Contrast, *p* = 0.53; CV2D6DifVarnc, *p* = 1.03; ATeta1, *p* = 0.073; CH3D6SumVarnc, *p* = 0.96; CN2S6Entropy, *p* = 0.64; CV5S6SumEntrp, *p* = 0.81). Four parameters (Perc01, ICC = 0.72; RNS6Fraction, ICC = 0.77; RNS6ShrtREmp, ICC = 0.41) were excluded from further processing due to the low ICC values. The ADC-based multivariate analysis showed a coefficient of determination of 0.78, an R^2^-adjusted of 0.61, and a multiple correlation coefficient of 0.84. Two parameters were excluded from the analysis due to high collinearity (as shown by a VIF > 104; CH1S6SumOfSqs and CV3S6Correlat). Of the remaining four parameters, none was demonstrated to be independently associated with the diagnosis of mucinous cysts. The T2-based and ADC map-based multivariate analysis results are displayed in Table 4. The diagnostic ability of the selected features to identify mucinous lesions is displayed in Table 5 and Figure 4. Texture maps based on feature distribution in T2WIs are shown in Figure 5. 

## 4. Discussion

Our results show that T2WI, DWI, and ADC maps-based signal intensity measurements were unsuccessful in differentiating mucinous from non-mucinous pancreatic cystic lesions. The limited utility of the ADC coefficients to differentiate between the two PLC entities was also concluded in most of the previously published studies (Table 5). In our study, the SIMs on DWI images had the highest variation (COV) among the observers. Although having the lowest variation coefficient (6.9%; 95% CI, 5.46–8.37%), the ADC values also held the lowest inter-rater agreement coefficient (k < 0.001). We were only able to identify one previous study [33] that quantified the Si on different b-value sequences as a differentiation tool for PCLs. In the study conducted by Mottola et al. [33], the DWI measurements were presented as cyst-to-pancreas SI ratios, which showed that mucinous tumors had lower DWI cyst-to-pancreas SI ratio, on both b = 750 s/mm^2^ (1.448 and 2.216, *p* = 0.013) and b = 1000 s/mm^2^ (1.094 and 1.941, *p* = 0.015). A summary of the main findings of previous studies that investigated the role of ADC values in differentiating mucinous from non-mucinous PLCs is displayed in Table 6.

Besides the variable and sometimes contradictory results, it is important to acknowledge that each of the previously published studies used different MRI protocols, therefore acquiring DWI images at different b-values. The b-value is a factor that mirrors the timing and strength of the gradients utilized to generate these sequences [39]. The DWI images are computed by turning diffusion-sensitizing gradients at various strengths [40], and the b-value is directly linked to the diffusion effects [39]. Our ADC measurements show that non-mucinous lesions had lower values than mucinous, which may seem counterintuitive at first glance. However, a deeper introspect into the MNPCs’ mucin characteristics may partially explain the variable SIMs obtained in this group and, thus, the non-significant statistical analysis results. Firstly, most MNPC aspirates are hypocellular (a feature that is expected to increase the ADC values) [2]. Secondly, The MNPCs’ pathological spectrum includes a wide variety of benign and malignant abnormalities, including non-neoplastic hyperplasia, adenoma, adenoma with severe atypia, and adenocarcinoma—each with its own microscopic features [41]. Moreover, the pancreatic epithelium produces different types of mucins in different stages of hyperplasia and malignant progression. Therefore, benign lesions tend to produce sulfated mucin, whereas malignant lesions tend to have neutral mucin or sialomucin [42]. The different types of mucin can also be expected to impact the SIMs. Boraschi et al. [37], who followed a similar design and also included multiple types of PLCs (IPMNs, pseudocysts, SCAs, and mucinous cystadenomas), observed analogous dynamics of the ADC coefficients. The authors [37] justified these results with the fact that SCAs have a multiseptate and multiloculate morphology (which includes fluid and solid components) and by their content, which includes glycogen-rich cells and proteinaceous fluid. In addition, the same study [37] demonstrated that the lowest ADC was associated with inflammatory lesions, which was attributed to the heterogeneity of the content found in pancreatitis-related collections, which can range from serous fluid to collections containing hemorrhagic or necrotic debris. What is important to consider is that in biologic tissue, the ADC values are influenced not only by the molecular diffusion of water, but also by the microcirculation of blood in the capillary network (perfusion characteristics). The perfusion effects typically influence the ADC coefficients when the map is computed from sequences with low b-values, which often increase registered ADC coefficients [35]. This perfusion effect may have influenced SCAs’ measured DWI and ADC values, both in the current and previous studies [35,36], because SCAs are typically hyper-vascular (hyperperfused tumors) [43]. It also needs to be acknowledged that cystic lesions can often be contaminated by bleeding, infection, or debris, which could further influence the measured SIs [44,45].

Our radiomics processing of MRI images showed that, based on T2WI, five features were independent predictors for MNPCs. The “Contrast” parameter (CH3D6Contrast) is a measure of intensity or gray-level variations between the reference pixel and its neighbor. In the visual perception of the real world, contrast is determined by the difference in the color and brightness of the object and other objects within the same field of view [46,47]. The contrast parameter increases its values when there is a large amount of variation within an ROI [45]. This parameter demonstrated higher values for MNPCs than for nMNPCs. Inverse Difference Moment (IVD) measures the local homogeneity of an image. It was previously demonstrated that the IVD’s weight value is the inverse of the Contrast weight [46,48]. This observation was validated by our findings, with the IVD parameter demonstrating higher values for non-mucinous lesions. All three of the selected Gray-Level Non-Uniformity (GLN) parameters displayed higher values for non-mucinous lesions. This metric increases when gray-level outliers dominate the histogram [49,50]. The GLN measures the variability of gray-level intensity values in the image, with a lower value indicating more homogeneity in intensity values. The interpretation of T2-based TA results may seem contradictory, with two independent parameters indicating a more heterogenous content for the MNPCs (CH3D6Contrast and CH5D6InvDfMom), while the three GLN parameters showed a more heterogenous content for the nMNPCs. However, both results can be true at the same time, considering the first two features were calculated through the Co-occurrence matrix and the GLN features were calculated through the Run-length matrix, each of them having their own computational method (i.e., looking at the same image through “different perspectives”). 

The ADC-based radiomics feature showed an overall lower sensitivity and higher specificity for the diagnosis of mucinous lesions (Table 5). However, none of them was able to independently predict the cysts’ content nature. Again, the Contrast parameter demonstrated higher values for MNPCs, as in the T2-based analysis. Wavelet energy quantifies the distribution of energy along the frequency axis over scale and orientation. Energy measures the local uniformity within an image. When the gray levels of an image are distributed under a constant or periodical form, energy becomes high [51]. Our analysis showed higher values for MNPCs than for nMNPCs. Again, the GLN feature showed higher values for non-mucinous lesions. Short- and long-run emphasis reflects the distribution of short or long homogeneous runs in an image. High values of the long-run emphasis indicate coarse surfaces [52], and in our study, they were observed for non-mucinous lesions. Apparently, the parameters followed the same dynamics, although they were independently computed from T2WI and ADC paps. However, the same “contradictory” results were observed when interpreting the absolute values recorded by the ADC-derived textures. Therefore, no direct assumptions regarding which histological feature influences which parameter can be made. Further studies are required to directly link the mucinous nature of the fluid with the values of one or more specific texture parameters.

We were able to identify several previously published studies that investigated the role of radiomics in PCLs’ diagnosis (Table 7). The two main directions that these studies investigated were tumor grading [19,20,21,22,23,24] and ductal adenocarcinoma survival [25,26,27,28,29]. A study conducted by Chen et al. [30] had a similar premise, investigating the CT-derived texture features’ ability to differentiate pancreatic serous cystadenomas from pancreatic mucinous cystadenomas. In this regard, the authors [30] built a combined model made of radiomic and conventional radiologic features, which was able to differentiate the two entities with 87.5–90% sensitivity and 82.4–84.6% specificity. Although this model outperformed the diagnostic ability of the classic radiologic features alone (75% sensitivity, 82.4% specificity), the authors [30] did not evaluate the radiomics features solely for diagnostic capabilities. Interestingly, all of the abovementioned radiomic studies only involved CT examinations [19,20,21,22,23,24,25,26,27,28,29,30]. For unknown reasons, the MRI-derived radiomic features’ role in differentiation PCLs has never been investigated, but considering that this technique could provide more heterogeneous images than CT, it is possible that more texture information could be extracted. 

The technical aspects should not be neglected. The decision to use multiple ROIs rather than incorporating collections into a larger volume of interest (VOI) may have influenced our findings. ADC fluctuations between slices in abdominal MRI were observed by Miquel et al. [53], who conclusively proved that these variations are much less likely to influence three-dimensional (3D) VOIs because any differences between (and within) slices are likely to be averaged over the large VOI. In- and interslice averaging does not occur with 2D ROIs, but this method has the advantage of higher reproducibility coefficients compared with the 3D analysis [40,53]. We agree that using VOIs would have offered a more “comprehensive” description of diffusion within the collections, while also accounting for within- and between-slice variations. Our approach, although possibly considered less accurate, is closer to the current use of ADC measurements in clinical practice, and is more straightforward and less time-consuming than VOI segmentation. By only selecting examinations that were performed on the same machine and processed on the same workstation, we were able to successfully counteract inter-scanner variability in ADC measurements as well as the effect of different post-processing software on ADC values [54]. This becomes particularly important when considering that previous research found up to 4% variability when using different MRI machines and up to 8% variability when processing ADC with different types of software [55]. Almost all of the previously published CT-based radiomic studies involving pancreatic lesions used manually-delineated ROIs [19,20,21,22,23,24,27,28,29]. One study used a VOI [25] and another used an automatic method for ROI delineation [26]. The single-slice ROI TA analysis that was used in the current study can be regarded as controversial [56]. Moreover, some of the included PCLs were small, and the top and bottom slices were often with artifacts. It has also not been clearly demonstrated that there is significant added value in radiomic analysis by undertaking multi-slice/volumetric analysis, as previous TA studies [57] that compared the two ROI definition techniques concluded that there is no significant difference between the two in selected applications. The spatial resolution, strength of the magnetic field, signal-to-noise ratio, and other acquisition parameters can impact the TA results [58]. We were able to counteract these influences by only choosing examinations that were conducted on the same machine under the same acquisition protocol. From both medical and management perspectives, it is of priority to evaluate as much information as possible from the standard MRI sequences, rather than administrating contrast or acquiring supplementary sequences [45]. In this regard, TA could become an important tool that could increase the confidence in the MRI diagnostic of PCLs, if further validated by larger prospective studies. 

Our study had several limitations. First, due to its retrospective design, it could have had selection bias. It remains debatable whether the inclusion of both pathologically confirmed and unconfirmed lesions could be regarded as a pitfall. However, not all PCLs required surgery, biopsy, or surgery per primam, and we only chose the lesions that underwent follow-up and had a clear imaging and clinical diagnosis. Similarly, the diagnosis was often assumed by imaging and/or follow-up without pathologic confirmation in most of the previously published studies with the same goal [13,33,35]. Moreover, being a retrospective study may have introduced verification bias regarding the patients’ clinical follow-up, which mainly depends on the status of the institution and referral hospital. The study population was also rather small, which was due to the strict inclusion criteria (especially the size criterion, which was necessary to provide enough surface for ROI placement and the difficulty of visualizing these lesions on ADC maps) and also due to the status of our institution. In particular, the SCA population was relatively small, a limitation also encountered in most similar previous studies [33,35,37]. The fact that one researcher (CC) was aware of the final diagnosis could also be considered a limitation. However, because at the time of the MRI examinations, some of the patients may have presented with multiple pancreatic lesions, this approach was necessary in order to only select documented lesions. After this step, that particular researcher was not involved in the processes of image segmentation, statistical analysis, or reporting the results. Splitting the cohort into training and validation (testing) groups is an important stage in the radiomics classification process. Approximately 70% of the acquired dataset is typically utilized for training, with the remaining samples being used to assess the features’ classification performance [59], which we were unable to perform due to the small cohort. Therefore, further studies are required to confirm our findings. In addition, our choice to use MaZda [60] as the imaging processing software may be viewed as outdated. Although various texture applications have been developed, only a few can provide built-in approaches for feature reduction and vector classification within an intuitive interface that may be utilized by non-image processing experts, such as medical physicians.

## 5. Conclusions

The MRI-based SIMs were unable to provide statistically significant results when comparing mucinous to non-mucinous PCLs. Radiomics features have the potential to augment the PLCs’ differential diagnosis, but future studies are required to investigate the extract histological substrate that influences PCLs’ textures.

## Figures and Tables

**Figure 1 healthcare-10-01039-f001:**
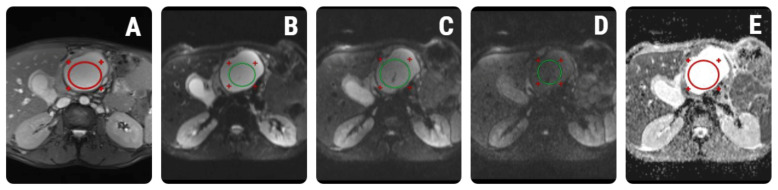
Quantitative signal intensity measurements using regions of interest (ROIs) placed on: (**A**) T2-weighted images (red ellipsoid); (**B**–**D**) three diffusion-weighted (DWI) sequences acquired at different b-values (green ellipsoids); and (**E**) apparent diffusion coefficient (ADC) maps (red ellipsoid). The examination belongs to a 43-year-old patient who developed pseudocysts after an episode of pancreatitis.

**Figure 2 healthcare-10-01039-f002:**
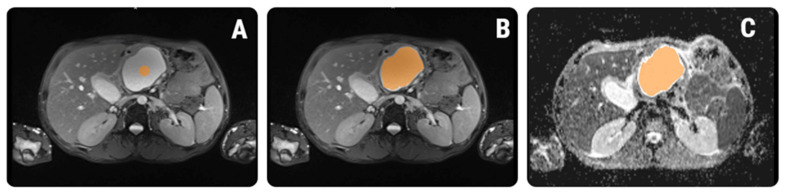
ROI definition within the texture analysis software. (**A**) A seed was placed on the T2-weighted image (orange round dot); (**B**) the software automatically grew the seed based on intensity and gradient coordinates (orange); (**C**) after extracting the features from the T2-weighted image, the ROI was transferred to the corresponding ADC map and manually adjusted (orange).

**Figure 3 healthcare-10-01039-f003:**
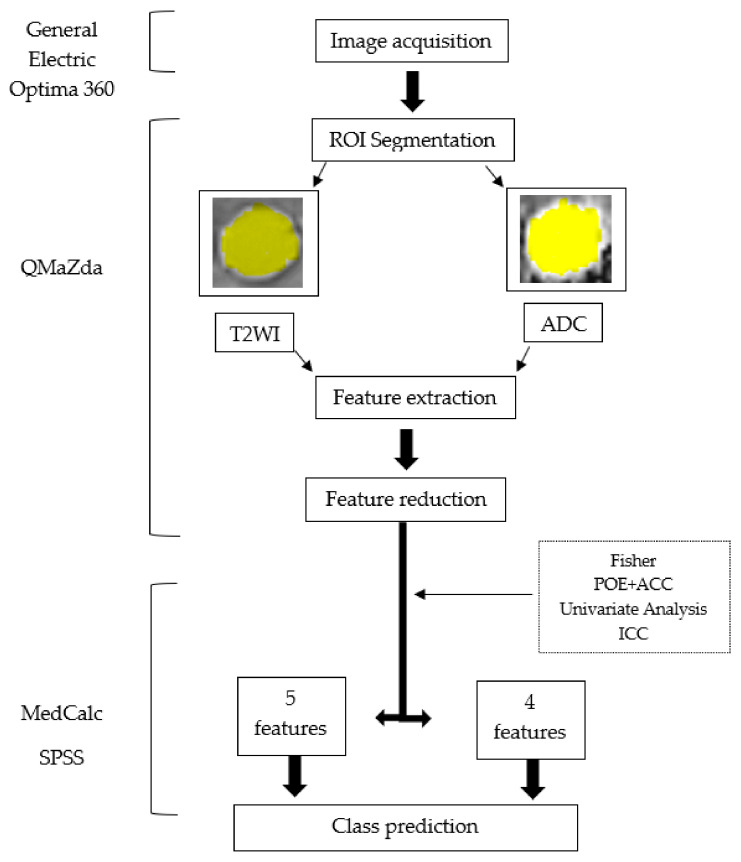
Radiomics workflow diagram. T2WI, T2-weighted images; ADC, apparent diffusion coefficient; ROI, region of interest.

**Figure 4 healthcare-10-01039-f004:**
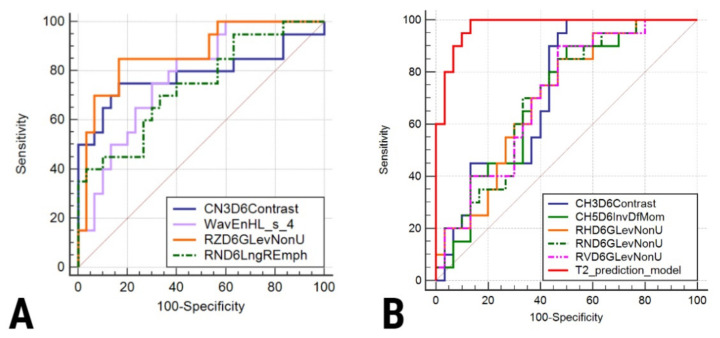
ROC curves of the parameters extracted from (**A**) ADC maps and (**B**) T2WI.

**Figure 5 healthcare-10-01039-f005:**
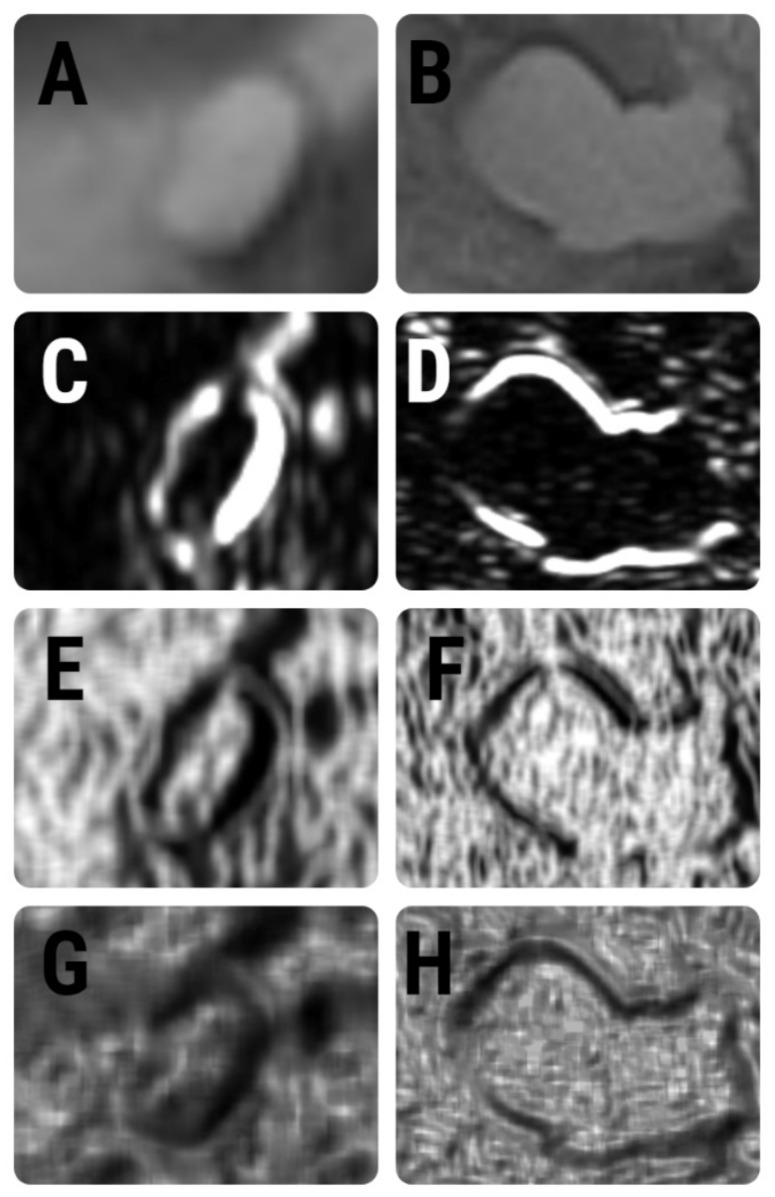
T2-weighted images of (**A**) side-branch IPMN and (**B**) pancreatic pseudocyst. Below each image are the maps that display the distribution of (**C**,**D**) CH3D6Contrast, (**E**,**F**) CH5D6InvDfMom, and (**G**,**H**) RHD6GLevNonU parameters.

**Table 1 healthcare-10-01039-t001:** Texture parameters.

Parameters (Texture Features)	Class	Computation Method	Computational Variations	*n*
Perc.01–99%, Skewness, Kurtosis, Variance, Mean	Histogram	-	-	5
GrNonZeros, percentage of pixels with nonzero gradient, GrMean, GrVariance, GrSkewness, GrKurtosis	AR	4 bits/pixel	-	5
GLevNonU, LngREmph, RLNonUni, ShrtREmp, Fraction	RLM	6 bits/pixel	4 directions	20
InvDfMom, SumAverg, SumVarnc, SumEntrp, Entropy, DifVarnc, DifEntrp, AngScMom, Contrast, Correlat, SumOfSqs	COM	6 bits/pixel; 5 between-pixel distances	4 directions	220
Teta 1–4, Sigma	ARM	-	-	5
WavEn	WT	5 scales	4 frequency bands	20

*n*, the total number of parameters extracted from each class; AR, absolute gradient; RLM, run length matrix; COM, co-occurrence matrix; ARM, auto-regressive model; WT, wavelet transformation; Mean, histogram’s mean; Variance, histogram’s variance; Skewness, histogram’s skewness; Kurtosis, histogram’s kurtosis; Perc.01–99%, 1st to 99th percentile; GrMean, absolute gradient mean; GrVariance, absolute gradient variance; GrSkewness, absolute gradient skewness; GrKurtosis, absolute gradient kurtosis; GrNonZeros, percentage of pixels with nonzero gradient; RLNonUni, run-length nonuniformity; GLevNonU, grey level nonuniformity; LngREmph, long-run emphasis; ShrtREmp, short-run emphasis; Fraction, the fraction of image in runs; AngScMom, angular second moment; Contrast, contrast; Correlat, correlation; SumOfSqs, the sum of squares; InvDfMom, inverse difference moment; SumAverg, sum average; SumVarnc, sum variance; SumEntrp, sum entropy; Entropy, entropy; DifVarnc, the difference of variance; DifEntrp, the difference of entropy; Teta 1–4, parameters θ1–θ14; Sigma, parameter σ; WavEn, wavelet energy.

**Table 2 healthcare-10-01039-t002:** Patients’ characteristics.

Groups	Entities	No Lesions/Patients	Mean Age (Years)	Sex (M/F)	Final Diagnosis		
					Clinical/ Imaging	Surgery	FNA
nMNPCs	pseudocyst	19/16	35.1	9/7	13	4	2
	WON	6/6	41.2	4/2	1	4	1
	SCA	4/4	47.6	0/4	3	1	-
	SPN	2/2	31.5	1/1	-	2	-
	SC	2/2	32.7	0/2	1	-	1
	cystic PNET	2/2	58.5	0/2	-	2	-
	cystic dAC	1/1	71	1/0	-	1	-
MNPCs	IPMN	18/16	62.4	3/13	-	12	1
	MCA	4/4	23	0/4	-	4	1
	MCC	1/1	58	1/0	-	2	-

No, number; nMNCPs, non-mucinous neoplastic pancreatic cysts; MNPCs, mucinous neoplastic pancreatic cysts; WON, walled-off necrosis; SCA, serous cystadenoma; SPN, solid pseudopapillary neoplasm; dAC, ductal adenocarcinoma; SC, simple cyst; PNET, pancreatic neuroendocrine tumor; FNA, fine needle aspiration; MCA, mucinous cystadenoma; MCC, mucinous cystadenocarcinoma; M/F, males/females.

**Table 3 healthcare-10-01039-t003:** Univariate analysis results and intraclass and coefficient of variations for the signal intensity measurements.

SIM	Non-Mucinous	Mucinous	*p*-Value	ICC	COV (%)
T2	434 (295.74–572.12)	451 (334.75–565)	0.69	0.79 (CI, 0.72–0.86)	9.6 (CI, 7.58–11.66)
b50	221 (151.85–263)	184.75 (119–261.25)	0.43	0.01 (CI, −0.02–0.05)	12.26 (CI, 9.65–14.93)
b400	73.5 (57.7–88.73)	60 (53–77.75)	0.21	0.009 (CI, −0.02–0.04)	16.2 (CI, 12.78–19.91)
b1000	22.85	26.75 (19.37–28.96)	0.21	−0.02 (CI, −0.04-−0.01)	32.38 (CI, 25.05–40.14)
ADC	2.82 (2.76–3.1) *	2.91 (2.56–2.99) *	0.98	<0.001	6.91 (CI, 5.46–8.37)

Bold values are statistically significant; SIM, signal intensity measurement; ICC, intraclass correlation coefficient (Kappa); COV, coefficient of variation from duplicate measurements; between the brackets, values corresponding to the interquartile range; CI, 95% confidence interval; * values are presented as number × 10^−3^ mm^2^/s.

**Table 4 healthcare-10-01039-t004:** Multivariate analysis results. Bold values indicate parameters that are able to independently predict mucinous cysts.

Texture Features	Coefficient	Std. Error	*p*	r_partial_
T2-based analysis				
CH3D6Contrast	−0.23	0.08	**0.01**	−0.4278
CH4D6DifVarnc	0.45	0.34	0.19	0.2337
CH5D6DifVarnc	−0.11	0.25	0.65	−0.08191
CH5D6InvDfMom	−3.82	1.62	**0.02**	−0.3896
CN4D6InvDfMom	10.2	9.7	0.3	0.1855
CN5D6InvDfMom	−6.15	7.14	0.39	−0.153
GD4Kurtosis	−0.001	0.001	0.51	−0.1111
Perc01	0.004	0.008	0.55	0.1065
Perc10	−0.01	0.007	0.13	−0.2662
RHD6GLevNonU	−0.04	0.01	**0.0007**	−0.5621
RND6GLevNonU	0.04	0.01	**0.0006**	0.567
RVD6GLevNonU	−0.04	0.01	**0.0012**	−0.5396
CN1S6SumAverg	0.32	0.008	0.21	0.4357
ADC-based analysis				
CN3D6Contrast	−0.24	0.356	0.078	0.125
WavEnHL_s_4	0.03	<0.01	0.065	0.286
RZD6GLevNonU	8.02	2.07	0.12	0.096
RND6LngREmph	1.06	0.8134	1.78	0.8134

Correlation, coefficient of determination R^2^; Std. Error, standard error; r_partial_, partial correlation coefficient; *p*, statistical value. Bold values are statistically significant.

**Table 5 healthcare-10-01039-t005:** Receiver-operating characteristics analysis results and the median values obtained by each parameter.

Texture Parameter	Mean Values		*p*-Value	AUC	J	Cut-Off	Sensitivity	Specificity
	nMNPCs	MNPCs						
T2-based analysis								
CH3D6Contrast	2.59	4.1	0.0016	0.72 (0.58–0.84)	0.5	>0.63	100 (83.2–100)	50 (31.3–68)
CH5D6InvDfMom	0.61	0.48	0.0062	0.7 (0.55–0.82)	0.41	≤0.66	90 (68.3–98.8)	50 (31.3–68.7)
RVD6GLevNonU	182.6	81.13	0.0039	0.71 (0.56–0.83)	0.43	≤125.35	90 (68.3–98.8)	53.33 (34.3–71.7)
RND6GLevNonU	243.7	107.2	0.005	0.7 (0.59–0.82)	0.38	≤149.24	85 (62.1–96.8)	53.33 (34.3–71.7)
RHD6GLevNonU	166.3	74.1	0.005	0.7 (0.55–0.82)	0.38	≤95.38	85 (62.1–96.8)	53.33 (34.3–71.7)
T2 prediction model	-	-	<0.0001	0.97 (0.88–0.99)	0.86	-	100 (83.2–100)	86.67 (69.3–96.2)
ADC-based analysis								
CN3D6Contrast	0.76	12.4	0.0001	0.8 (0.66–0.9)	0.65	>0.94	75 (50.9–91.3)	90 (73.5–97.9)
WavEnHL_s_4	126.64	338.7	<0.0001	0.77 (0.63–0.88)	0.45	>141.1	85 (62.1–96.8)	60 (40.6–77.3)
RZD6GLevNonU	91.6	41.87	<0.0001	0.87 (0.74–0.94)	0.68	≤50.77	85 (62.1–96.8)	83.33 (65.3–94.4)
RND6LngREmph	191.35	46.7	0.0009	0.74 (0.57–0.85)	0.36	≤3.56	40 (19.1–63.9)	96.67 (82.8–99.9)

*p*-value, significance level for the ROC analysis; J, Youden index; sensitivity and specificity are reported as percentages; between the brackets, the values corresponding to the 95% confidence intervals.

**Table 6 healthcare-10-01039-t006:** Previous studies that used the apparent diffusion coefficient (ADC) values for the differentiation of mucinous from serous pancreatic cystic lesions.

Author, Year	Measurements		*p*-Value	Utility	b-Values
	ADCm	ADCs			
Pozzessere et al., 2017 [32]	3.26	2.86	<0.001	cut-off > 3; Se, 84–88%; Sp, 66–72%;	50, 750;
Mottola et al., 2012 [33]	2.6	2.14	0.013	not reported	0, 250, 500, 750, 1000
Sandrasegaran et al., 2012 [34]	2.99	2.31	0.12	not reported	50, 400, 800;
Irie et al., 2001 [35]	2.8	3.2 *	no statistically significant result (*p*-value not reported);		30, 300, 900;
Yamashita et al., 1998 [36]	2.7	3.2 *	no statistically significant result (*p*-value not reported);		30, 300;
Current study	2.91	2.82	0.98	not reported	50, 400, 800;

*p*-value, statistical significance value; ADC, apparent diffusion coefficient; ADC values are expressed as number x × 10^−3^ mm^2^/sec; b-values are expressed in s/mm^2^; ADCm, reported mean ADC values for mucinous cysts; ADCs, reported mean ADC values for serous cysts; * mean ADC values for the serous cyst groups composed only of pseudocysts. Bold values are statistically significant.

**Table 7 healthcare-10-01039-t007:** Previously published radiomics studies involving pancreatic lesions.

Aim	Lesion	Author, Year
Tumor grading	IPMN	Permuth et al., 2016 [19]
		Attiyeh et al., 2019 [20]
		Chakraborty et al., 2018 [21]
	Ductal adenocarcinoma	Cassinotto et al., 2017 [22]
	Neuroendocrine tumor	Canellas et al., 2018 [23]
		Choi et al., 2018 [24]
Survival	Ductal adenocarcinoma	Hyun et al., 2016 [25]
		Eilaghi et al., 2017 [26]
		Chakraborty et al., 2017 [27]
		Sandrasegaran et al., 2019 [28]
		Attiyeh et al., 2018 [29]

IPMN, intraductal papillary mucinous neoplasm.

## Data Availability

Not applicable.

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
