# Peer review of "Quantitative MRI of Pancreatic Cystic Lesions: A New Diagnostic Approach"

_healthcare, 2022, doi:10.3390/healthcare10061039_

Round 1

Reviewer 1 Report

  1. Key words are reflect the focus of the manuscript, but should be simplified.
  2. The first author self-cited 11 times. Some of the citations have nothing to do with the topic of the article.

Reviewer 2 Report

In this retrospective study, authors investigated the role of quantitative MRI in the distinction between mucinous and non-mucinous pancreatic cystic lesions, focusing on SIM and radiomics analysis.

Overall, the article is well organized and structured, and results are clearly described in tables and figures.

  • Introduction should be summarized
  • Lines 131-132: authors define the acronym SIM as signal intensity measurement, but, in the text, they still use SI measurement (e.g. line 321, Table 2) for the same value.
  • In the Methods, authors should specify the Tesla of the MRI machine
  • A Table with patients’ characteristics could be added to have an overview about the included population
  • Conclusions should be improved, especially regarding the absence of statically significance of SIM
  • Authors should check the maximum number of keywords according to journal’s guidelines

Round 2

Reviewer 1 Report

Dear Author;
Thank you for response.

Author Response

Point 1: Dear Author; Thank you for response.

Answer 1: Dear Reviewer, thank you for the good observations and your effort in reviewing this manuscript. With gratitude, The Authors